# The Impact of Exercise Training on Physical Activity among Elderly Women in the Community: A Pilot Study

**DOI:** 10.3390/healthcare11182601

**Published:** 2023-09-21

**Authors:** You-Ying Lii, Yao-Chung Tai, Hung-Yi Wang, I-Chen Yeh, Yen-Chun Chiu, Chieh-Yi Hou, Feng-Hua Tsai

**Affiliations:** 1Physical Education Office, National Kaohsiung University of Science and Technology, Kaohsiung 81157, Taiwan; jylii@nkust.edu.tw (Y.-Y.L.); yicfe@nkust.edu.tw (I.-C.Y.); 2Marine Leisure Management, National Kaohsiung University of Science and Technology, Kaohsiung 81157, Taiwan; yctai@nkust.edu.tw; 3Department of Sports Technology and Leisure Management, Center for General Education, I-Shou University, Kaohsiung 84001, Taiwan; 4Department of Orthopaedic Surgery, E-Da Hospital, Kaohsiung 82445, Taiwan; ed104259@edah.org.tw; 5Department of Physical Medicine and Rehabilitation, E-DA Cancer Hospital, Kaohsiung 82445, Taiwan; ed100699@edah.org.tw; 6Center for Physical and Health Education, Nation Sun Yat-Sen University, Kaohsiung 804, Taiwan; shawn63211@mail.nsysu.edu.tw

**Keywords:** exercise training, physical activity, functional fitness

## Abstract

According to a survey conducted by the Taiwanese government, the elderly spend most of their time watching TV for their daily leisure activities, and most do not pant or sweat during exercise. Relevant studies have shown that physical activity has benefits and importance for the physical functions of the elderly. The objective of this study was to investigate the effects of exercise training on the functional fitness of elderly females in the community. The women subjects were from the community of Kaohsiung City, Taiwan. In total, 34 females were randomly divided into an intervention group and a control group. The ages of the subjects ranged from 65 to 80 years old, with an average age of 75.13. The experimental group continued their physical activity intervention for 20 weeks. The control group had no training plan. The results of the study showed that after 20 weeks of intervention, participants in the intervention group experienced improvements in back grasping, right-hand grip strength, sitting to standing, right hip flexion, right knee extension, right ankle dorsiflexion, right sitting forward extension balance, and sitting back around objects. Six-minute walking distances also showed a significant difference in all cases. The results demonstrated that the 20-week physical activity program intervention used in this study can assist in improving back grasping, right-hand grip strength, sitting to standing, right-hand sitting posture, forward balance, sitting back around objects, and six-minute walking distance among older women in the community. In summary, we recommend a moderate-intensity physical activity exercise program for older women in the community.

## 1. Introduction

At the end of August 2018, Taiwan’s Ministry of the Interior registered 8,164,676 households with elderly people over the age of 65. Elderly households compose 14.56% of total households in the country. This indicates that Taiwan is turning into an elderly society. Additionally, Kaohsiung City ranked third in the number of elderly households, with 326,635 households [1]. In the future, with the rapid increase in the elderly population, the dependency ratio of elderly households will increase from 2022 to 2050 [2]. Lung respiratory efficiency, lung ventilation, gas exchange function, and lung capacity decrease by about 40% to 50% by the age of 70 [3], and maximum muscle strength and muscle mass decrease by about 25% between the ages of 40 and 65 [4]. The American College of Sports Medicine (ACSM) noted that aging will lead to a decline in cardiorespiratory endurance and muscle fitness, which will affect physical function and athletic performance ability. A decrease in cardiorespiratory endurance and muscle fitness will determine whether the elderly will suffer from disease, disability, and death in the future [5]. Diseases associated with increased mortality in the elderly include cardiovascular disease, coronary heart disease, chronic disease, and cancer, and are closely related to physical activity [6]. Research shows that older adults with better physical fitness or higher levels of physical activity have significantly lower cardiovascular disease risk factors than older adults with average or poorer physical fitness [5]. The 2017 Survey of the Condition of the Elderly noted that the rate of chronic diseases in women is increasing faster than that of men and exceeds that of men aged 75 to 79 [7]. According to the 75-year-old frailty assessment, women lose 10.56% to 11.36% of their weight and lower limb functions. Functional weakness was found to increase by 21.37–40.39%, while energy decreased by 4.95–6.16%; the other three indicators of weakness were also higher than those of men. Additionally, 80.73% of those at the age of 65 reported leisure activities of watching TV, followed by 52.89% reporting outdoor fitness and exercise and 46.91% reporting conversation, making tea, and singing. The 2013 Sports City Survey [8] noted that 82.1% of Chinese people participate in sports, including 84.7% of men and 79.5% of women. In addition, 58.6% of Chinese people engage in regular exercise at the ages of 65–69, as do 51.9% of Chinese people over 70 years old. Many studies reported that a correlation existed between aging and exercise habits. The average number of exercise sessions per week among Chinese people is 3.37; for Chinese individuals over 70 years old, this number is 5.30, and for those 65–69 years old, the number is 4.81. The average duration of every exercise time for people aged 70 is 60.7 min. In terms of exercise intensity, only 25.4% of people over the age of 70 sweat and pant during exercise. The American College of Sports Medicine (ACSM) and the American Heart Association (AHA) recommend moderate-intensity aerobic activities, muscle flexibility, and balance exercises to promote physical activity in the elderly to reduce prolonged sitting behavior and falling risk [9]. Regardless of whether older adults have diseases or not, increasing their physical activity levels will enhance their physical functions. Moreover, a dynamic lifestyle is negatively correlated with chronic diseases [10,11,12]. Functional physical fitness is defined to include components such as body composition, muscular fitness, flexibility, cardiovascular endurance, and balance [13,14]. Based on the above-mentioned problems, such as the decline in physical activity among the elderly in the community due to disability and aging, the purpose of this study is to explore the current level of physical activity among female elderly people in the community and plan exercise programs based on different levels of physical activity. After the intervention of the physical activity exercise program, we reassessed the effectiveness of the program for elderly females. Finally, the research results provide a reference for the planning of the physical activity exercise programs for elderly females in the community.

## 2. Materials and Methods

### 2.1. Participants

A total of 34 elderly females from Hunei District, Kaohsiung City, were included in the study, while individuals with significant mental, cardiac, neurological, respiratory, and musculoskeletal diseases were excluded. The design of the study was conducted using experimental research methods. Participants were randomly divided into an intervention group and a control group, with 17 people in each. The intervention group (EG) received 20 weeks of moderate-intensity physical activity intervention that was performed 3 times a week, with 90 min of physical activity intervention each time, and the control group (CG) did not have any exercise plan. No behavior change was required. Four people in EG had colds, stomachaches, and shoulder pain, and they asked to leave the program a total of 6 times. However, they all returned to the exercise course and completed the entire program. Consecutive three-time misses are excluded. Recruitment began in May 2020 and was completed in December 2020. The testing site is at the community activity center.

### 2.2. Intervention Program

#### Research Design Process

The physical activity plan designed in this study is based on and modified from References of Hoeger and Hoeger [15] and Nelson [9]. The intervention group exercised 3 days a week with moderate intensity, scoring 5–6 points for Rating of Perceived Exertion (RPE), which corresponds to feeling a little breathless and having difficulty speaking, with 90 min of physical activity. The exercise patterns included a warm-up, joint stretching activity, muscle strength activity, dynamic balance, and aerobic training, as well as brain and leisure activities. In addition, the elements of panting and extremity exercising after exercising for more than 30 min were included. This exercise program is designed by members with backgrounds in orthopedics, attending surgeons, physical therapists, and national physical fitness instructors, aiming to promote brain health and physical activity. For more information, see Table 1 (exercise type) and Table 2 (physical activity program).

### 2.3. Materials

We used a multicomponent instrument to measure functional physical fitness performance.

#### 2.3.1. Back Scratch Test

This test measures the flexibility of shoulder rotation movement, which corresponds to the distance between the right shoulder and the left shoulder and the hands touching the back. The female norm at 75 years old is −1.3~12.7 cm [14].

#### 2.3.2. Right-Hand Grip Strength Test

This test measures the value obtained by the subject when using the maximum strength of the hand to grasp the gripper. The female norm at 75 years old is 46 pounds [16].

#### 2.3.3. Sit-Up Chair Stand Test

This test measures the number of times the subject continuously moves from a sitting posture to a standing posture in 30 s. The female norm at 75 years old is 10–15 times [14].

#### 2.3.4. Chair Sit and Reach Test

This test measures the distance that the subject can reach forward with her arm raised horizontally when the subject independently maintains balance. For females over 75 years old, a result less than 26.7 cm indicates a difference [14].

#### 2.3.5. Eight-Foot Up-and-Go Test

This test measures the time that it takes for the subject to walk to 2.44 m, return, and sit back down. The female norm at 75 years old is 5.2~7.4 s [14].

#### 2.3.6. Six-Minute Walking Distance

The subject walked back and forth in a straight area of 20 m for 6 min; then, the total length of walking was recorded. The female norm at 75 years old is 400~535 m [14].

### 2.4. Design and Data Analysis

This study applied both a pre-experimental design and a one-group pre-test–post-test experimental design. The dependent variables were the impact of functional fitness on performance and the measured impact of participants’ motivation on satisfaction. The exercises were conducted by a trained instructor. The participants completed a pre-test that covered their basic personal data, health status, and functional fitness status one week before the exercise course intervention. The G-power software (Ver. 3.1) was used to calculate the sample size, resulting in an estimated size of 16 participants per group (power = 0.8, alpha = 0.05, and effect size = 0.7).

The following statistical analyses were conducted using SPSS for Windows 22.0 for the data collected in this study. Basic data were analyzed using descriptive statistics to explore the distribution of basic variables among the participants. A two-way mixed design ANOVA was used to examine the effects of two independent variables on a dependent variable. It is interested in studying the effects of an intervention on participants in two different groups (the intervention group and the control group). In addition, the paired Sample *t*-test was used to analyze performance for functional fitness scores before and after the course intervention. A *p*-value less than 0.05 was regarded as statistically significant.

## 3. Results

The compliance rate of all 34 elderly females in our study was 100%. The basic data of the subjects in the intervention group and control group are shown in Table 3. The statistical results indicate no significant difference in age or BMI between the two groups. The values for flexibility, upper and lower limb muscle strength, balance, agility, and cardiopulmonary fitness among the subjects in the intervention group and control group before and after an exercise intervention of 20 weeks are shown in Table 4. There was no significant difference in flexibility, upper and lower limb muscle strength, balance, agility, or cardiorespiratory fitness between the intervention group and control group before exercise intervention. The intervention group participants’ flexibility (back scratch), lower limb muscle strength (chair stand), balance (chair sit and reach), and cardiorespiratory fitness (six-minute walking distance) after 20 weeks of exercise intervention were significantly different from those of the control group. The presence or absence of exercise intervention (intervention group and control group) and the measurement phase (pre- and post-test) were analyzed using 2 × 2 two-factor mixed design variance analysis. The statistical test results showed that flexibility (back scratch), upper limb muscle strength (hand grip strength), the interaction of muscle strength in the lower limbs (chair stand), balance (chair sit and reach), agility (sitting back around the object), and cardiorespiratory fitness (six-minute walking distance) reached a significant level (*p* < 0.05). The results of the two-factor mixed design reached a significant level, regardless of the intervention. The measurement stages were further analyzed using a paired *t*-test. The results of the two-factor mixed design variance analysis are shown in Table 4. Back scratching, right-hand grip strength, sitting to standing, right-hand sitting forward balance, sitting around objects, and six minutes of walking reached a significant level (*p* < 0.05). The analysis results are shown in Table 5. The results show that there were significant changes in flexibility (back scratch) (*t* = 7.75), upper limb muscle strength (hand grip strength) (*t* = −12.35), lower limb muscle strength (chair stand) (*t* = −10.10), balance (chair sit and reach) (*t* = −10.75), agility (sitting back around the object) (*t* = 8.24), and cardiopulmonary fitness (six-minute walking distance) (*t* = −8.39). In the control group, we recorded the test after 20 weeks. The pre-test and post-test results did not present a significant difference in the control group.

## 4. Discussion

### 4.1. Exploring the Current Level of Physical Activity

The results found that back scratch, hand grip strength, chair stand, chair sit and reach, sitting back around the object, and six-minute walking distance were all below the standard levels due to poor joint mobility, lack of muscle strength, limb alignment deviation (genu valgum and genu varum), etc. This result indicates that the subjects were all unhealthy before the intervention. In this study, a portion of elderly females in the community were suffering from chronic diseases and debilitating indicators, and their leisure activities were mostly watching TV. These elderly females reported not panting or sweating during exercise, which is consistent with the above potential problems. Relevant studies have noted [14,17] that a decline in functional physical fitness can lead to physical function limitations and ultimately reduce physical function and cause disabilities. As one’s age increases, the performance of cognitive functions will also gradually decline [18,19].

### 4.2. Improvements in Physical Activity Abilities among Elderly Females in the Community after the Intervention

The results of the study showed that after 20 weeks of intervention (exercise type and physical activity program), participants in the intervention group experienced significant improvements in back scratch, hand grip strength, chair stand, chair sit and reach, sitting back around the object, and six-minute walking distance. In the control group, the test results did not indicate a significant difference in statistical verification, and the performance of functional fitness was worse. The intervention of the activity program was found to help improve the upper lower extremity muscles, respiratory muscles, abdominal muscles, gluteus muscles, endurance, coordination, and dynamic balance for elderly females in the community. Related studies [5,9,15,17,20,21] noted that appropriate enhancement of aerobic training and resistance training can significantly improve muscle strength, flexibility, dynamic balance and agility, cardiorespiratory fitness, and cognitive performance. Short-term physical activity intervention can enhance functional fitness performance but may still not surpass the normative standards [14]. This could be due to significant age disparities within the sample group, poor baseline capabilities during pre-testing, and the effects of gradual aging. We look forward to devising a long-term physical activity plan in the future.

Empirical research indicates that aerobic endurance exercise and physical resistance exercise training (3 times a week for about 30 to 70 min each time, lasting 8 to 12 weeks with moderate intensity) can improve flexibility, muscle strength, timing, and dynamic balance, six-minute walking endurance, and other abilities [22,23,24,25]. Based on the above research results, after the 20-week exercise program intervention, the indexes of back scratch, hand grip strength, chair stand, chair sit and reach, and sitting back around the objects all improved. There were significant improvements observed in the six-minute walking distance evaluation items, showing that physical activity program training can help improve the functional fitness (flexibility, muscle strength, endurance, and balance) performance of elderly females in the community and help them avoid suffering from chronic diseases, debilitating indicators, and decreased physical function.

## 5. Conclusions

We found that the functional fitness performance of elderly females in a community of Kaohsiung City, Taiwan, was inadequate. After 20 weeks of physical activity program intervention, the indexes of back scratch, hand grip strength, chair stand, chair sit and reach, sitting back around the object, and six-minute walking distance improved significantly. This result is in line with the above-mentioned literature on flexibility. The results of the research on muscle strength, dynamic balance, agility, and cardiorespiratory endurance performance indicate that the 20-week physical activity program is a popular program for elderly females in the community and can improve the performance of functional physical fitness.

This research was limited by time, funding, manpower, and sample size. Thus, the results cannot be generalized to female elderly people in the whole city or country. However, the moderate-intensity exercise during the activities universally produced heavy breathing and reduced the level of speaking, consciously reminding the subjects of their efforts. In addition, this study did not investigate the leisure time of the subjects, so it is impossible to know whether the subjects increased their exercise training. The above research limitations are considerations for readers seeking to design a similar exercise scheme. If community courses are being promoted, we recommend that the history of physical illness, physical activity, and exercise habits among participants be investigated during the first class. Then, a functional fitness assessment along with testing and exercise guidance should be carried out so that elderly females in the community can understand their functional fitness and current fitness performance abilities. After the course, 30 min should be reserved for female senior citizens in the community to learn exercises independently, according to the advantages and disadvantages of their personal data. Ultimately, we hope that this study will provide a reference for physical activity exercise plans for elderly females in the community.

## Figures and Tables

**Table 1 healthcare-11-02601-t001:** Exercise type.

Item	Exercise Action	Benefits
Respiratory Exercise	Stand up and lift the basketball with both hands up and down	U/E muscles and respiratory muscles
Fingers add force to ball	Sit and stand with two fingertips pressing against each other on a basketball	U/E muscles and respiratory muscles
Knees add force to ball	Sit and stand with both knees pinching the ball	L/E muscles and abdominal muscles
Ankles add force to ball	Sit and stand with both ankles straight and hold the basketball flat	L/E muscles and abdominal muscles
Toe touch the ball	Sit on the ground and touch the ball continuously with the left and right feet	L/E muscles and abdominal muscles
Cross touch the ball	Sit on the spot and straddle the ball with the left foot and right foot while kicking	L/E muscles, abdominal muscles, and coordination
Square ball	Sit and stand with the left foot (right foot); then, step on the ball and draw a square clockwise (counterclockwise)	L/E muscles, abdominal muscles, and coordination
Thera-band	Sit and stand with an elastic band around the ankle of the left foot (right foot), move the elastic band on the ankle of the right foot (left foot), and perform resistance exercises on the three sides of the front, back, and left	U/E and L/E muscles, abdominal muscles, gluteus muscle, and coordination
Bounce the ball	Sit and stand in place with the left hand (right hand) and bounce the ball continuously; then, stand in place with the left hand (right hand) and bounce the ball continuously	U/E muscles, endurance, and coordination
Hand dribble	Move the basketball back and forth continuously around the twenty-meter marker tube	U/E muscles, endurance, coordination, and dynamic balance
Tiptoe	Raise one’s toes on the basketball with both hands	U/E and L/E muscles, endurance, coordination, and dynamic balance
Walking	Go back and forth continuously and briskly around the twenty-meter marker tube	L/E muscles, endurance, and dynamic balance
Alternate bounce	Sitting on the spot, using the left and right hands (right and left hands), alternately bouncing the ball continuously, with new instructions given randomly and instantly	U/E muscles and coordination
Finger exercise	Sit and stand with the left and right fingers raised at the same time (random numbers)	U/E muscles
Pass and catch	While sitting on the ground, throw the ball and bounce it toward one’s teammates. First, pass with the left hand (feet) to connect with the left hand (feet); then, pass from the left to connect with the right, pass with the right to connect with the right, and pass with the right to connect with the left	U/E and L/E muscles and coordination

**Table 2 healthcare-11-02601-t002:** Physical activity program.

Sessions					
Exercises	4 wk	8 wk	12 wk	16 wk	20 wk
Breathing exercise (S × R)	2 × 10	2 × 12	2 × 15	3 × 10	3 × 12
Fingers add force to ball (S × R)	2 × 10	2 × 12	2 × 15	3 × 10	3 × 12
Knees add force to ball (S × R)	2 × 10	2 × 12	2 × 15	3 × 10	3 × 12
Ankles add force to ball (S × R)	2 × 10	2 × 12	2 × 15	3 × 10	3 × 12
Toe touch the ball (S × R)	2 × 10	2 × 12	2 × 15	3 × 10	3 × 12
Cross touch the ball (S × R)	2 × 10	2 × 12	2 × 15	3 × 10	3 × 12
Square ball (S × R)	2 × 10	2 × 12	2 × 15	3 × 10	3 × 12
Thera-band (S × R)	2 × 10	2 × 12	2 × 15	3 × 10	3 × 12
Bounce the ball (S × M)	2 × 1 m	3 × 1 m	2 × 2 m	3 × 2 m	2 × 3 m
Hand dribble (S × M)	2 × 1 m	3 × 1 m	2 × 2 m	3 × 2 m	2 × 3 m
Tiptoe (S × M)	2 × 1 m	3 × 1 m	2 × 2 m	3 × 2 m	2 × 3 m
Walking (M)	3 m	6 m	6 m	9 m	9 m
Alternate bounce (S)	2	2	3	3	4
Finger exercise (S × R)	2 × 3	2 × 6	3 × 3	3 × 6	4 × 3
Pass and catch (S × M × D)	1 × 5 m	2 × 5 m	3 × 5 m	2 × 3 m	2 × 3 m

S × R: sets × repetitions; S × S: sets × second; S × M: sets × minute; R × D: repetitions × distance; S: sets; M: minute; and M × D: minute × distance.

**Table 3 healthcare-11-02601-t003:** Survey results for the basic data of participants.

Variable	Intervention Group (n = 17)	Control Group (n = 17)
Age (years old)		
Mean ± standard errors	74.73 ± 4.03	75.52 ± 4.12
BMI (kg/m^2^)		
Mean ± standard errors	26.20 ± 3.62	27.29 ± 2.86

**Table 4 healthcare-11-02601-t004:** Two-way mixed design ANOVA for participants in the intervention group and control group.

Items	SS	DF	MS	F Test	*p*-Value
Back scratch (cm)					
Training	162.13	1	162.13	1.14	0.29
Timing	606.12	1	606.12	78.28 *	<0.01
Variable test	430.02	1	430.02	55.54 *	<0.01
Hand grip strength (pounds)					
Training	144.13	1	144.13	1.02	0.32
Timing	259.41	1	259.41	148.24 *	<0.01
Variable test	181.19	1	181.19	103.54 *	<0.01
Chair stand (times)					
Training	24.72	1	24.72	2.63	0.11
Timing	67.35	1	67.35	137.22 *	<0.01
Variable test	44.49	1	44.49	90.64 *	<0.01
Chair sit and reach (cm)					
Training	84.94	1	84.94	3.69	0.06
Timing	311.18	1	311.18	148.62 *	<0.01
Variable test	204.77	1	204.77	97.80 *	<0.01
Eight-foot up-and-go test (seconds)					
Training	3.94	1	3.94	0.29	0.59
Timing	44.22	1	44.22	99.38 *	<0.01
Variable test	29.89	1	29.89	67.17 *	<0.01
Six-min walking distance (meter)					
Training	11,079.77	1	11,079.77	3.87	0.06
Timing	25,010.85	1	25,010.85	101.69 *	<0.01
Variable test	15,967.12	1	15,967.12	64.92 *	<0.01

Note: * *p* < 0.01.

**Table 5 healthcare-11-02601-t005:** Numerical analysis of participants in the intervention group and control group before and after exercise intervention.

Items	Intervention Group (n = 17)				Control Group (n = 17)			
	Before Intervention	20 Weeks	*t*-Value	*p*-Value	Before Intervention	20 Weeks	*t*-Value	*p*-Value
Back scratch (cm)	22.35 ± 11.59	13.53 ± 7.19	7.75 *	<0.01	20.41 ± 8.29	21.65 ± 6.74	−1.70	0.11
Hand grip strength (pounds)	39.94 ± 9.19	45.76 ± 9.28	−12.35 *	<0.01	40.29 ± 7.84	39.59 ± 7.35	1.62	0.12
Chair stand (times)	8.88 ± 2.52	11.94 ± 2.25	−10.10 *	<0.01	9.29 ± 1.96	9.12 ± 2.12	1.14	0.27
Chair sit and reach (cm)	21.06 ± 4.28	27.65 ± 2.76	−10.75 *	<0.01	22.29 ± 3.84	21.94 ± 3.09	1.03	0.32
Eight-foot up-and-go test (seconds)	11.88 ± 3.16	9.43 ± 2.08	8.24 *	<0.01	11.03 ± 2.58	11.23 ± 2.65	−1.59	0.13
Six-min walking distance (meter)	353.47 ± 41.54	413.47 ± 21.98	−8.39 *	<0.01	358.59 ± 46.81	357.29 ± 42.69	0.50	0.62

Note: * *p* < 0.01.

## Data Availability

Data are contained within the article.

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
