# Peer review of "The Impact of Exercise Training on Physical Activity among Elderly Women in the Community: A Pilot Study"

_healthcare, 2023, doi:10.3390/healthcare11182601_

Round 1

Reviewer 1 Report

- The overall procedure including the rationale, the methodology, the result and the conclusion described in this study is very articulate to understand. In addition, there is no need for a proof-reading. 

- However, it seems that the possibility of replication needs to be resolved. The researchers of this study need to show or tell differentiated results from the previous studies.    

- Theoretical and practical rationale needs to be reinforced. For instance, the ratinale needs to answer the questions such as "Is it necessary to find out the kind, the level or the intensity of physical activity for the female elder although ACSM and AHA already recommend them?" or "Is there societal, cultural or any other contextual reason to update them?"

Author Response

Responses:

  1. Thanks for your suggestions. We have made the necessary changes “Related studies [5,9,15,17,20,21] noted that appropriate enhancement of aerobic training and resistance training can significantly improve muscle strength, flexibility, dynamic balance and agility, cardiorespiratory fitness, and cognitive performance. Short-term physical activity intervention can enhance functional fitness performance, but still may not surpass the normative standards [14]. This could be due to significant age disparities within the sample group, poor baseline capabilities during pre-testing, and the effects of gradual aging. We look forward to devising a long-term physical activity plan in the future.”. ( Lines 253-261)
  2. Thanks for your suggestions. We have made the necessary changes “The American College of Sports Medicine (ACSM) and the American Heart Association (AHA) recommend moderate-intensity aerobic activities, muscle flexibility, and balance exercises to promote physical activity in the elderly to reduce prolonged sitting behavior and falling risk [9].” ( Lines 81-85)
  3. Thanks for your suggestions. We believe that these factors, which were societal, cultural, or any other contextual reasons, can be further considered in future research. However, due to limitations in the scope of the study, these factors cannot be included for exploration in this research.

Reviewer 2 Report

The title states "physical activity" but its "functional fitness" within the text.

Lines 32-33, "older people" should be "older women".

Lines 41-45, no clear rationale between functional fitness and the disesases mentioned. Needs proper citation.

Line 46-47,  is misleading, cardiorespiratory causal effect was not determine in this study.

Line 48, this was not established in this study.

Lines 50-52, misleading.

Should define what is functional fitness in context of this study.

Literature is under reviewed. Hence no clear gap of knowledge could be deduced. 

Need strong justification on why females were selected. 

See;

Marsala, M. J., Belfry, S., Orange, J. B., & Christie, A. D. (2023). Sex-Related Differences in Functional Fitness Outcomes in Older Adults. Journal of aging and physical activity31(4), 556–567.

Wróblewska, Z., Chmielewski, J. P., Florek-Łuszczki, M., Nowak-Starz, G., Wojciechowska, M., & Wróblewska, I. M. (2023). Assessment of functional capacity of the elderly. Annals of agricultural and environmental medicine : AAEM30(1), 156–163.

Cunha, P. M., Nunes, J. P., Werneck, A. O., Ribeiro, A. S., da Silva Machado, D. G., Kassiano, W., Costa, B. D. V., Cyrino, L. T., Antunes, M., Kunevaliki, G., Tomeleri, C. M., Fernandes, R. R., Junior, P. S., Teixeira, D. C., Venturini, D., Barbosa, D. S., Qian, Y. U., Herold, F., Zou, L., Mayhew, J. L., … Cyrino, E. S. (2023). Effect of Resistance Exercise Orders on Health Parameters in Trained Older Women: A Randomized Crossover Trial. Medicine and science in sports and exercise55(1), 119–132.

Results

What is the compliance rate? Any drop outs?

Table 3 is not very scientific. 

Tables 4 and 5 are hard to follow. Not sure what was measured/units.

The discussion should be in light of current body of knowledge as there many studies including RCTs circa 2022 and 2023.

Author Response

Responses:

  1. Thanks for your suggestions. This study refers to the assessment of the effectiveness of the intervention through pre-tests and post-tests of functional physical fitness after implementing the physical activity plan. Therefore, the term "functional fitness" will appear in the text.
  2. Thanks for your suggestions. We have revised "older people" to "older women". (Lines 36-37)
  3. Thanks for your suggestions. We have made the necessary changes “Research shows that older adults with better physical fitness or higher levels of physical activity have significantly lower cardiovascular disease risk factors than older adults with average or poorer physical fitness [5].”. (Lines 60-63)
  4. Thanks for your suggestions. We have made the necessary changes “Regardless of whether older adults have diseases or not, increasing their physical activity levels will enhance their physical functions. Moreover, a dynamic lifestyle is negatively correlated with chronic diseases. [10, 11, 12].”. (Lines 85-88)
  5. Thanks for your suggestions. We have made the necessary changes “Functional physical fitness is defined to include components such as body composition, muscular fitness, flexibility, cardiovascular endurance, and balance [13, 14].”. (Lines 88-90)
  6. Thanks for your suggestions. We have made the necessary changes “The 2017 Survey of the Condition of the Elderly noted that the rate of chronic diseases in women is increasing faster than that of men and exceeds that of men aged 75 to 79 [7]. According to the 75-year-old frailty assessment, women lose 10.56% to 11.36% of their weight and lower limb functions. Functional weakness was found to increase by 21.37%-40.39%, while energy decreased by 4.95%-6.16%; the other three indicators of weakness were also higher than those of men. Additionally, 80.73% of those at the age of 65 reported leisure activities of watching TV, followed by 52.89% reporting outdoor fitness and exercise, and 46.91% reporting conversation, making tea, and singing. The 2013 Sports City Survey [8] noted that 82.1% of Chinese people participate in sports, including 84.7% of men and 79.5% of women.”. (Lines 63-74)
  7. Thanks for your suggestions. We have made the necessary changes “The compliance rate of all 34 elderly females in our study was 100%.”. (Line 185)
  8. Thanks for your suggestions. We have revised Table 3.
  9. Thanks for your suggestions. We have explicitly identified the units of measurement for each variable in Table 3, Table 4 and Table 5.
  10. Thanks for your suggestions. We have made the necessary changes “Related studies [5,9,15,17,20,21] noted that appropriate enhancement of aerobic training and resistance training can significantly improve muscle strength, flexibility, dynamic balance and agility, cardiorespiratory fitness, and cognitive performance. Short-term physical activity intervention can enhance functional fitness performance, but still may not surpass the normative standards [14]. This could be due to significant age disparities within the sample group, poor baseline capabilities during pre-testing, and the effects of gradual aging. We look forward to devising a long-term physical activity plan in the future.”. (Lines 253-261)

Reviewer 3 Report

This research holds significant implications in reinforcing the understanding of exercise benefits for elderly women. Some revisions are recommended below: 

  • Line 63-64: The cited report does not appear to establish a definitive causal relationship between aging and exercise habits. It would be more accurate to characterize these factors as "related." 

  • Line 88-89: Please provide additional details regarding the temporary absence of participants and its potential impact on exercise adaptation. For example, were the four people who briefly left the program from the same group (EG or CG)? How many times did each participant leave?  

  • Table 3: The variables presented in the exercise habit survey and exercise intensity survey seem to represent "participant counts," but the units are not clearly stated. The authors should explicitly identify the units of measurement for each variable in this table. 

  • Table 4: The inclusion of the Mandarin characters "項目" appears redundant and could be omitted. These characters literally translate to "items," a term already stated. 

  • Table 4: Including the P-values within the table is required to enhance the clarity of the findings. 

  • Table 5: Similarly, displaying the P-values within the table for Table 5 would contribute to the overall transparency of the results. 

Author Response

Responses:

  1. Thanks for your suggestions. We have made the necessary changes “Many studies reported that a correlation existed between aging and exercise habits.”. (Lines 76-77)
  2. Thanks for your suggestions. We have made the necessary changes “Four people in EG had colds, stomachaches, and shoulder pain, and they asked to leave the program a total of 6 times. However, they all returned to the exercise course and completed the entire program.”. (Lines 109-112)
  3. Thanks for your suggestions. We have explicitly identified the units of measurement for each variable in Table 3, Table 4 and Table 5.
  4. Thanks for your suggestions. We have omitted the inclusion of the Mandarin characters "項目" in Table 4.
  5. Thanks for your suggestions. We presented the P-values within the table for Table 4 and Table 5.

Reviewer 4 Report

Dear Atuhors, please see the attached file

 The use of English is fine. Minor editing of English language required. However, I am not a native English speaker. Thus, I suggest you the contribution of a native speaker to ensure an right use of English. 

Author Response

Responses:

  1. Thanks for your suggestions. We have made the necessary changes “This study was approved by E-Da Hospital Human Testing Committee reviewed and passed (EMRP10109N).”. ( Lines 314-315)
  2. Thanks for your suggestions. We have made the necessary changes “The design of the study was conducted using experimental research methods. Participants were randomly divided into an intervention group and a control group with 17 people in each. The intervention group (EG) received 20 weeks of moderate-intensity physical activity intervention, 3 times a week, with 90 minutes of physical activity intervention each time, and the control group (CG) did not have any exercise plan. No behavior change is required. Four people in EG had colds, stomachaches, and shoulder pain, and they asked to leave the program a total of 6 times. However, they all returned to the exercise course and completed the entire program. Consecutive three times misses are excluded. Recruitment began in May 2020 and was completed in December 2020. The testing site is at the community activity center.”. ( Lines 103-114)
  3. Thanks for your suggestions. We have made the necessary changes “A total of 34 elderly females from Hunei District, Kaohsiung City, were included in the study, while individuals with significant mental, cardiac, neurological, respiratory, and musculoskeletal diseases were excluded.” ( Lines 101-103)
  4. Thanks for your suggestions. We have made the necessary changes “The physical activity plan designed in this study is based and modified from References Hoeger & Hoeger [15] and Nelson [9]. The intervention group exercised 3 days a week with moderate intensity scoring 5-6 points for Rating of Perceived Exertion (RPE), which corresponds to feeling a little breathless and having difficulty speaking, with 90 minutes of physical activity. The exercise patterns included a warm-up, joint stretching activity, muscle strength activity, dynamic balance, and aerobic training, as well as brain and leisure activities. In addition, the elements of panting and extremity exercising after exercising for more than 30 minutes. This exercise program is designed by members with backgrounds in orthopedics attending surgeons, physical therapists, and national physical fitness instructors, aiming to promote brain health and physical activity. For more information, see Table 1 (exercise type) and Table 2 (physical activity program).” ( Lines 117-134)
  5. Thank you for your suggestions. We have incorporated the 6 new references into the introduction section (Lines 253-261). Additionally, int he discussion section, we have made the necessary changes ” Related studies [5,9,15,17,20,21] noted that appropriate enhancement of aerobic training and resistance training can significantly improve muscle strength, flexibility, dynamic balance and agility, cardiorespiratory fitness, and cognitive performance. Short-term physical activity intervention can enhance functional fitness performance, but still may not surpass the normative standards [14]. This could be due to significant age disparities within the sample group, poor baseline capabilities during pre-testing, and the effects of gradual aging. We look forward to devising a long-term physical activity plan in the future.” ( Lines 253-261)

Round 2

Reviewer 1 Report

Researchers' authentic idea was added enough to increase readers' interest.  

Author Response

  1. Thanks for your suggestions and your affirmation of our research. We will continue to work hard to perfectly present this research.

Reviewer 4 Report

Dear Authors,

Thank you very much for your effort in improving the manuscript. Now it is much better. Great job!

English is fine an minor editing of English language is required. However, I am not a native English speaker. Therefore, I suggest you the consultancy of a native English author to be sure of a perfect Use of English. 

Author Response

Thanks for your suggestions and your affirmation of our research. We will continue to work hard to perfectly present this research.